# Association Study on Nutrition in the First Year of Life and Molar-Incisor Hypomineralization (MIH)—Results from the GINIplus and LISA Birth Cohort Studies

**DOI:** 10.3390/ijerph182111411

**Published:** 2021-10-29

**Authors:** Yeganeh Khazaei, Carla P. Harris, Joachim Heinrich, Marie Standl, Jan Kühnisch

**Affiliations:** 1Department of Conservative Dentistry and Periodontology, University Hospital of Ludwig-Maximilians University of Munich, 80336 Munich, Germany; yeganekhazaei@gmail.com; 2Institute for Medical Information Processing, Biometry and Epidemiology, Ludwig-Maximilians University of Munich, 81377 Munich, Germany; 3Helmholtz Centre Munich-German Research Center for Environmental Health, Institute of Epidemiology, 85764 Neuherberg, Germany; carla.harris@helmholtz-muenchen.de (C.P.H.); marie.standl@helmholtz-muenchen.de (M.S.); 4Department of Paediatrics, Dr. von Hauner Children’s Hospital, Ludwig-Maximilians University of Munich, 80337 Munich, Germany; 5Institute and Clinic for Occupational, Social and Environmental Medicine, University Hospital of Ludwig-Maximilians University of Munich, 80336 Munich, Germany; Joachim.Heinrich@med.uni-muenchen.de; 6Allergy and Lung Health Unit, Melbourne School of Population and Global Health, The University of Melbourne, Carlton, Melbourne, VIC 3053, Australia

**Keywords:** molar-incisor hypomineralization, developmental enamel defects, early nutrition, exclusive breastfeeding, birth cohort study, epidemiology, aetiology research

## Abstract

Molar-incisor hypomineralization (MIH) is a condition with specific clinical presentation whose etiology to date still remains unknown. This study prospectively investigated the association between nutrition during the 1st year of life and the presence of MIH in the permanent dentition. Data from 1070 10-year-old children from two prospective birth cohort studies were included. Information on exclusive breastfeeding (EBF) and introduction of 48 food items into the child’s diet was assessed at 4-, 6-, and 12-month time-points. Food diversity was defined according to the number of food items or food groups introduced into the child’s diet and then subsequent categorization into low-, middle- and high-diversity groups was performed. MIH was scored in the permanent dentition at age of 10 years. The statistical analysis included logistic and Poisson hurdle regression models adjusted for potential confounders. EBF, food item and food group diversity at 4-, 6-, 12-month time-points were found to be non-significant in most of the categories for the development of MIH. However, significantly higher odds for the presence of MIH were found for certain categories. Despite the limitation of this study, such as arbitrary cut-offs for categorization of food items, the results of this study suggest the lack of an association between early nutrition in the first year of life and MIH in the permanent dentition.

## 1. Introduction

Molar-incisor hypomineralization (MIH) is a prevalent dental finding in children and adolescent populations [1,2,3] and is considered a multifactorial condition, meaning it is influenced by lifestyle factors, environmental exposures, and genetic preconditions [4]. Potential related factors have been extensively investigated, including antibiotic treatment during early childhood [5,6,7], and medical problems such as respiratory diseases, fever, maternal stress, maternal smoking and alcohol consumption during pregnancy [7,8,9], however, the etiology of MIH has not yet been clarified [8,10,11].

Long-term prospective epidemiological studies that evaluated the effect of nutrition on MIH, especially during the first year of life, are scarce. The potential association between MIH and early nutrition—in terms of breastfeeding—was first highlighted by Alaluusua et al. [12,13]. The authors suggested that a long duration of breastfeeding might be associated with an increase in MIH in developing teeth. This association was assumed to be linked to environmental contaminants in human milk [10,12,14]. Later, Laisi et al. [15] documented that there was no significant correlation between breastfeeding and MIH. However, the results seem to be controversial and need to be further investigated [5,6,11,16]. On the other hand, exposing children’s developing teeth to a wide diversity of food may lead to a higher incidence of MIH in the future due to hidden environmental pollutants in food interacting with the body, as well as with cell development [17]. The effect of environmental toxins on MIH development has been a topic of interest [18]; however, the impact of feeding practice during the first months of life and food diversity at 6 and 12 months on the occurrence of MIH has not been investigated before. Therefore, this study aimed to assess the association between key variables of early diet and the presence of MIH in two prospective birth cohort studies. Our null hypothesis was that there is no association between early diet in terms of breastfeeding, food item diversity and food group diversity and MIH development.

## 2. Materials and Methods

*Study population.* The study population is based on GINIplus (German Infant Nutritional Intervention plus environmental and genetic influences on allergy development) and LISA (Influence of Lifestyle-Related Factors on the Immune System and the Development of Allergies in Childhood) birth cohort studies. The longitudinal study design, populations and dental examinations have been described extensively elsewhere [2,19,20]. In GINIplus, a total of 5991 healthy full-term newborns were recruited at birth between 1995 and 1998 in the study centers Munich and Wesel, while in LISA, 3097 healthy, full-term newborns were enrolled in the study from 1998 to 1999 in the study centers Munich, Leipzig, Wesel, and Bad Honnef.

All dental examinations were conducted during the 10-year follow-up in the Munich study center by trained and calibrated examiners (mean of age = 10.20, standard deviation of age = 0.25) [20,21]. Informed written informed consent was obtained from at least one legal guardian, as well as from the participants themselves before the follow-up examination (Figure 1). This standard procedure was consented from the ethical committee. The study was approved by the ethics committee of the Bavarian Board of Physicians (No. 05100 for GINIplus and No. 07098 for LISA).

*Dental examination.* Details of the study team including examiners, methodology of the dental examination, training procedures, and calibration data were described earlier [20,21]. Prior to the dental examination, all participants brushed their teeth. The standardized examination equipment included a dental mirror, a blunt CPI probe (CP-11.5B6, Hu-Friedy, Chicago, IL, USA), and a halogen lamp (Ri-Magic, Rudolf Riester GmbH, Jungingen, Germany). Each child was investigated at the designated appointment for the presence of caries (dmf/DMF index; WHO 1997) and MIH by one calibrated dentist. All permanent teeth with demarcated opacities, enamel breakdowns and atypical restorations or extractions on at least one index tooth were categorized as MIH according to the criteria of the European Academy of Paediatric Dentistry (EAPD) [1,2]. Hypomineralized lesions with a diameter of <1 mm were not recorded [1]. Other enamel defects, e.g., hypoplasia, fluorosis (diffuse opacities), amelogenesis imperfecta and dentinogenesis imperfecta, were clearly distinguished from MIH and were not recorded. MIH-associated defects or restorations were not scored in the DMF index. Children with at least one affected first permanent molar were classified as having MIH, and those with demarcated enamel hypomineralization on their first permanent molars and incisors were categorized as M + IH [1,2,22]. Children with at least one enamel hypomineralization on a permanent tooth were categorized as HT ≥ 1 [2,22]. Otherwise, subjects with no demarcated opacities, enamel breakdowns or atypical restorations were scored as being free of MIH or enamel hypomineralization.

*Definition of exclusive breastfeeding (EBF) and food diversity*. Information on exclusive breastfeeding (EBF) and the introduction of 48 food items into the child’s diet was collected by using self-administered questionnaires that were completed by the parents during the first 4 months and at the 6 and 12-month follow-ups. EBF during the first 4 months of life was defined as a dichotomous variable: “exclusively breastfed” for each of the first 4 months of life or not. Non-exclusive breastfeeding (non-EBF) was defined as being mixed breast and bottle fed or exclusively bottle fed for at least one of the first four months of life.

Food diversity was defined according to Markevych et al. [23]. The 48 food items were grouped into 8 categories: (1) vegetables (avocado, cauliflower, beans, broccoli, peas, cucumbers, carrots, potatoes, white cabbage, turnip cabbage, lenses, celery, asparagus, spinach, tomatoes, onion, vegetable juices); (2) fruit (apples, pineapples, apricots, bananas, pears, strawberries, peaches, citrus fruit, fruit juices); (3) cereal (bread/pretzels/rolls, cookies/cakes/rusk, rolled oats, muesli, millet, cornmeal/corn starch, wheat semolina/starch, noodles, rice/rice starch, spelt); (4) meat (poultry, lamb, veal/beef, pork, sausages); (5) egg; (6) dairy products (cow milk/cream, yoghurt/quark/cheese); (7) fish; and (8) other (nuts, soy products, cocoa/chocolate). Participants with missing information on at least 9 out of the 48 food items were excluded, which amounted to 20% of the questionnaire and were considered as randomly occurring. For the remaining participants with sporadic blanks, the missing values were replaced with “not yet introduced”. Food diversity was defined by summing the number of food items (48 food items) or food groups (8 food groups) introduced into the child’s diet at age 6 months and 12 months. Afterwards, subsequent categorization into low-, middle- and high-diversity groups was performed. This was done by dividing the specific ordered distribution in the complete study population into three parts, separately in food item diversity and food group diversity. Low-, middle- and high-diversity groups were identified specifically for each category, as shown in Table 1. The numbers in the parentheses are indicating the range of the number of food items and groups and were used in the models, while the low group of each category was set as a reference group. In addition, the association between single food items and MIH was investigated.

*Statistical Analysis*. For the descriptive analysis, frequencies and percentages are presented for categorical variables, and mean values with standard deviations (SDs) are presented for continuous variables. Mean values and SDs as well as crude associations were calculated for each of the defined MIH phenotypes (HT ≥ 1, MIH and M + IH). The associations of breastfeeding and diversity of food items and groups in the first 6 and 12 months with the presence of MIH were analyzed using logistic regression analysis. Here, the middle- and high-diversity groups were compared to the reference group (low-diversity group) with respect to MIH development. Odds ratios (ORs) were computed with corresponding 95% confidence intervals (95% CIs) and *p*-values. Since the prevalence of hypomineralization was low, the crude number of HTs showed a positively skewed distribution with a large stack of zero counts for children without hypomineralization. Therefore, a hurdle regression model was used to account for this zero-inflated distribution [24]. The first part of this model used logistic regression for the probability of a nonzero count, which refers to hypomineralization prevalence. Odds ratios (ORs) were calculated. The second part of the model used negative binomial regression for the mean count among the subjects with a nonzero count, which reflects hypomineralization severity. Relative risks (RRs) were determined. The results are presented as ORs, RRs, 95% confidence intervals (95% CIs), and corresponding *p*-values (P) [25].

In sensitivity analyses, the association of single food items was analyzed separately with the outcome variables that showed significant associations in the main analyses aiming to investigate whether the food diversity effect is driven by specific food items or groups. These single food items included each of 8 food groups and 48 food items, limited to those being consumed by more than 5% and less than 95% of the study population. These analyses were adjusted for multiple testing using Bonferroni correction. The adjusted alpha level was 0.001 (0.05/43). Further sensitivity analyses were performed using different cut-offs for missing values of 0 and 19 instead of 9. In addition, investigation of breast feeding as a continuous variable was also performed, with values ranging from 0 to 6 months. All models were adjusted for sex, age, body mass index (BMI) at 10 years, parental education level, maternal atopy, and study group (study was either GINIplus observation arm, GINIplus intervention arm, or LISA). These confounders were a priori selected based on existing literature [23]. A statistical comparison was considered significant if the two-sided *p*-value was <0.05. This study had a sufficient sample size with 80% power at a significance level of α = 5% to detect a 17% difference in the proportions of the population who developed MIH between the low and high food item diversity groups. Analyses were performed using the statistical software R 3.3.2. [26]. Poisson hurdle regression models were used, as implemented in the R package “pscl” [27].

## 3. Results

The study population (*N* = 1070) characteristics are shown in Table 1. The prevalence of developmental dental defects ranged from 9.2% in the M + IH group to 13.7% in the MIH group; 24.9% of the 10-year-old children had at least one hypomineralization in the permanent dentition. A total of 73.6% of infants were exclusively breastfed (EBF) for the first 4 months. At 6 months of age, the low-diversity groups of children consuming 0–1 food items and food groups comprised 41.9% and 47.9% of the population, respectively. These numbers in the low-diversity group at 12 months of age increased to 4–27 food items and 2–6 food groups (Table 1).

Adjusted associations between dietary variables and the presence of HT ≥ 1, MIH and M + IH are shown in Table 2. In most of the categories, non-significant results were recorded. No significant associations were observed between EBF and HT ≥ 1, MIH, or M + IH. However, children in the middle group of food item diversity—introduction to 28–33 food items in the first 12 months of life—showed significantly higher odds of developing MIH and M + IH than those in the low-diversity group (<27 food items) (Table 2). In addition to the logistic regression models, the number of HTs was analysed in relation to EBF and food diversity using Poisson hurdle regression models (Table 2). Non-significant associations were observed for the first part of the model; however, in the second part of the model, a significant RR was observed for food item diversity in the middle diversity group at 12 months of life (RR = 1.62, 95% CI (1.18–2.22), *p* < 0.01) compared to that in the low diversity group (Table 2).

The results of the sensitivity analyses showed a non-significant association between single food items—8 food groups and 48 food items—and dental outcomes after adjusting for multiple testing (Table 3). Furthermore, the results of the sensitivity analyses of different missing value cut-offs and breast feeding as a continuous variable remained the same; therefore, the data are not shown. There was no significant association between the confounders and dental outcomes.

## 4. Discussion

This association study investigated the relationship between MIH and early diet during the first 6 and 12 months of life in 10-year-olds. The results indicated mostly non-significant associations between early diet and MIH, meaning our null hypothesis could not be rejected. However, the findings of this study suggested increased odds for MIH according to specific food categories.

The documented results of the relationship between EBF during the first 4 months of life and later presence of developmental dental defects in terms of MIH indicate a nonsignificant association. Previously, Alaluusua et al. [10] suggested that prolonged breastfeeding (more than 9–10 months) may lead to defects in enamel; however, it is possible that a duration to 4 months of EBF is not long enough to detect any visible effect, such in the present study. This is in accordance with data from a later study by a Finnish work group [15]. It should be considered that EBF for 6 months is proposed by the World Health Organization (WHO) mainly due to its beneficial immunological, developmental, and nutritional effects on infants [28]. Moreover, dichotomous EBF variable during the first 4 months of life was inspired by the latest position paper published by the European Society for Paediatric Gastroenterology, Hepatology, and Nutrition (ESPGHAN), suggesting at least 4 months of EBF as a desirable goal [29]. It has been discussed that those infants introduced to complementary foods before the age of 4 months have higher rates of illness or other health problems than other infants [30]. Based on what we know of the multifactorial nature of MIH, it may be expected to observe an association between the effect of EBF during the first 4 months of life and solid food introduction during the first year of life and MIH development.

The influence of early diet on MIH was investigated in this study for the first time. Previous studies showed a relationship between the timing of solid food introduction/solid food diversity and allergic diseases; therefore, we speculated that such an association may exist between early diet and MIH development. The main findings from the conducted statistical analyses were mostly non-significant associations. Nevertheless, for 12-month-old children with a more diverse early diet, the probability of developing MIH-related developmental defects was increased. In detail, a significant association between MIH or M + IH development was found for children that were introduced to 28–33 food items in the first year of life compared to the children in the low-diversity group, with a maximum of 27 food items. This result is in line with the increased RRs of hypomineralization severity when applying Poisson hurdle models. One may speculate that children with higher risks of MIH consumed higher-diversity diets; however, this effect was not consistent with the results of other food categories. We assume this inconsistency could be due to different ranges in the food diversity groups. Data transformation was required due to the highly skewed distribution, and to achieve sufficient sample size in each group, the categorization into three diversity groups for both 6 and 12 months was applied. Non-equal number of ranges were distributed between food categories (Table 2), which might explain the significant results in the middle-diversity group and non-significant results in the high-diversity group in the same category. Therefore, the results should be interpreted with caution. By further investigation into single food items in the sensitivity analyses, we observed that some food items, such as apricot, wheat/starch and dairies, were found to increase the odds of MIH and M + IH and the severity of hypomineralization. However, after adjustment for multiple testing, none of these associations remained significant. Additionally, these findings showed no clear pattern. Therefore, these results should not be overstated. In the existing literature, no previous study investigated such an association; hence, we cannot make any concrete statements about it.

A key strength of this study is that it was a prospectively designed longitudinal cohort study with sufficient power that examined the possible correlation between MIH at the age of 10 years and nutritional information from the first year of life, e.g., EBF vs. non-EBF practices and food diversity. The analysis of food diversity in relation to MIH was not conducted before; therefore, this study may contribute to the ongoing scientific discussion on MIH aetiology. Using two different statistical regression models based on data characteristics improved the strength of this study. Other advantages of this study included clinical recording of MIH according to the most recently published EAPD criteria, strict data selection, inclusion of potential confounders, and thorough data exploration. These advantages gave us the power to look thoroughly into the MIH distribution pattern and how to identify MIH and M + IH in the permanent dentition [11].

A few factors may have limited our ability to reach comprehensive conclusions. We should take into account the fact that the cut-offs of this study were arbitrary. Therefore, the categories and the associated ORs, RRs, and *p*-values are based upon subjective interpretations. However, our results were robust when testing different cut-offs. Furthermore, residual confounding cannot be ruled out due to the observational nature of our study. Therefore, unobserved factors affecting both parental decisions on infant feeding and solid food introduction as well as MIH development might at least partly explain the observed associations. Additionally, we did not have exact information on the amount or frequency of the children’s consumption of different food items or groups, as the main focus was on the timing of solid food introduction when the questionnaire was administered to the parents. Furthermore, it must be considered that the current dietary recommendations for children have changed since the late 1990s, thus signifying the need for future studies. Another limitation of this study is participation bias and nonrandom loss to follow-up due to the long follow-up duration, with a higher participation rate among those included in the intervention arm of the GINIplus study. The characteristics of the participants included in the analysis may differ in regard to higher socioeconomic status; hence, generalizability might be limited. An issue that must be considered is the reliability of parental-based questionnaires, which may lead to misclassification. While these findings deepen our understanding of the issue, they also highlight the gap in knowledge and scientific evidence. Thus, we strongly recommend further investigation via relevant large-scale prospectively designed clinical/observational studies to evaluate the effects of different factors, such as the frequency of consumption and the amount and composition of macronutrients, such as carbohydrates, fat, and protein, in children’s diets, on the occurrence of MIH.

## 5. Conclusions

With the limitations discussed, based on the present results, no clear associations between nutrition in the first year of life and the appearance of MIH in permanent teeth later were found.

## Figures and Tables

**Figure 1 ijerph-18-11411-f001:**
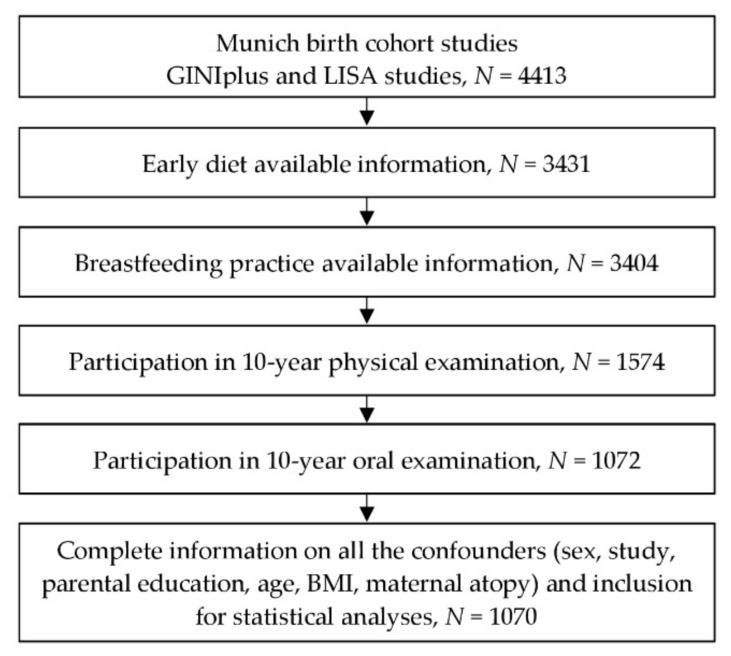
Flow chart of study participants from the time point of examination.

**Table 1 ijerph-18-11411-t001:** Crude associations between potential risk factors for caries and the MIH definitions used.

Proportion for Each Variable	Category	Analysis Population	HT ≥ 1 ^§^ %		MIH ^¶^ %		M + IH ^¶¶^ %	
		*N* (%)	No	Yes	No	Yes	No	Yes
Study	Overall	1072 ^¶¶¶^ (100)	75.1	24.9	86.3	13.7	90.8	9.2
	GINI-Observation	252 (23.5)	15.7	7.7	20.2	3.3	21.5	2.0
	GINI-Intervention	372 (34.7)	24.3	10.4	29.3	5.4	31.0	3.8
	LISA	448 (41.8)	35.1	6.8	36.8	5.1	38.3	3.4
Sex	Boys	545 (50.8)	38.2	12.7	43.1	7.7	45.8	5.0
	Girls	527 (49.2)	37.0	12.1	43.2	6.0	45.1	4.1
Parental education ^†^	Low ≤10 y	236 (22.9)	16.6	5.5	20.0	2.1	21.0	1.4
	High >10 y	835 (77.0)	58.6	19.3	66.3	11.7	70.2	7.7
	Missing	1 (0.1)						
Maternal atopy ^††^	Yes	543 (50.7)	37.4	13.2	43.9	6.7	46.3	4.4
	No	529 (49.3)	37.8	11.6	42.3	7.1	44.6	4.8
Breastfeeding during the first 4 months ^‡^	Nonexclusive	276 (25.7)	18.4	7.4	22.7	3.1	23.9	2.1
	Exclusive	789 (73.6)	56.7	17.5	63.6	10.6	66.9	7.1
	Missing	7 (0.7)						
Food item diversity during the first 6 months	Low [0,1]	449 (41.9)	30.8	11.1	36.3	5.6	38.1	3.8
	Middle [2–8]	332 (31.0)	24.3	6.7	25.9	5.0	27.2	3.7
	High [9–48]	291 (27.1)	20.1	7.0	24.1	3.1	25.6	1.6
Food group diversity during the first 6 months	Low [0–1]	513 (47.9)	35.4	12.4	41.8	6.1	43.9	4.0
	Middle [2–3]	291 (27.1)	20.7	6.4	23.1	4.0	24.2	2.9
	High [4–8]	268 (25.0)	19.0	6.0	21.4	3.7	22.8	2.2
Food item diversity during the first 12 months	Low [4–27]	376 (35.1)	26.3	8.8	31.4	3.6	32.6	2.4
	Middle [28–33]	349 (32.6)	24.5	8.0	27.0	5.5	28.8	3.7
	High [34–48]	347 (32.4)	24.3	8.0	27.8	4.6	29.4	3.0
Food group diversity during the first 12 months	Low [2–6]	564 (52.6)	38.8	13.8	45.7	6.9	47.9	4.7
	Middle [7]	257 (24.0)	18.0	6.0	20.6	3.3	21.7	2.2
	High [8]	251 (23.4)	18.4	5.0	20.0	3.4	21.2	2.2

^†^ Defined as highest educational level of mother or father. ^††^ Defined as mother reported to have hay fever, allergic rhinitis, allergic conjunctivitis, atopic eczema, or physician-diagnosed asthma. ^‡^ Defined as exclusively breastfeeding without any formula feeding within the first 4 months of life. ^§^ Enamel hypomineralization on at least one permanent tooth (HT ≥ 1). ^¶^ Enamel hypomineralization on at least one first permanent molar (MIH). ^¶¶^ Enamel hypomineralization on at least one first permanent molar and incisor (M + IH). ^¶¶¶^ Statistical models used complete information on all confounders, *N* = 1070.

**Table 2 ijerph-18-11411-t002:** Adjusted odds ratios/relative risks with 95% confidence intervals and *p*-values obtained from logistic regression analysis and Poisson hurdle regression analysis of exclusive breastfeeding during the first 4 months of life and diversity of food items and groups during the first 6 and 12 months of life and dental defect outcomes at 10 years of life. All models were adjusted for sex, study, age, BMI, maternal atopy, and parental education.

Exposure Variables		Category	*N* (%)	Logistic Regression Model						Hurdle Regression Model			
				HT ≥ 1		MIH		M + IH		HT ≥ 1 Prevalence		HT ≥ 1 Severity	
				OR (95% CI)	*p*	OR (95% CI)	*p*	OR (95% CI)	*p*	OR (95% CI)	*p*	RR (95%CI)	*p*
Breastfeeding during the first 4 months (*N* = 1062)		Non-EBF	274 (25.9)	1		1		1		1		1	
		EBF	788 (74.1)	0.93 (0.68–1.3)	0.68	1.19 (0.78–1.86)	0.42	1.30 (0.78–2.25)	0.33	1.06 (0.78–1.44)	0.71	1.18 (0.86–1.63)	0.31
Food Item Diversity	6 months (*N* = 1070)	Low [0–1]	449 (41.9)	1		1		1		1		1	
		Middle [2–8]	331 (31.0)	0.74 (0.52–1.04)	0.09	1.32 (0.88–1.98)	0.18	1.42 (0.89–2.27)	0.14	0.92 (0.67–1.25)	0.58	1.15 (0.86–1.55)	0.35
		High [9–48]	290 (27.1)	0.87 (0.61–1.24)	0.44	0.89 (0.55–1.42)	0.63	0.67 (0.36–1.20)	0.19	0.78 (0.56–1.09)	0.14	0.81 (0.57–1.14)	0.23
	12 months (*N* = 1070)	Low [4–27]	375 (35.1)	1		1		1		1		1	
		Middle [28–33]	348 (32.6)	1.01 (0.72–1.43)	0.94	**1.82 (1.18–2.85)**	**<0.01**	**1.81 (1.08–3.09)**	**0.03**	1.27 (0.93–1.75)	0.13	**1.62 (1.18–2.22)**	**<0.01**
		High [34–48]	347 (32.4)	1.01 (0.71–1.43)	0.95	1.53 (0.97–2.43)	0.07	1.48 (0.86–2.59)	0.16	1.11 (0.80–1.54)	0.52	1.25 (0.89–1.76)	0.19
Food Group Diversity	6 months (*N* = 1070)	Low [0–1]	512 (47.9)	1		1		1		1		1	
		Middle [2–3]	291 (27.1)	0.84 (0.59–1.18)	0.32	1.22 (0.79–1.85)	0.36	1.33 (0.81–2.18)	0.25	0.95 (0.69–1.30)	0.75	1.16 (0.85–1.57)	0.36
		High [4–8]	267 (25.0)	0.82 (0.57–1.17)	0.28	1.28 (0.82–1.99)	0.27	1.19 (0.68–2.02)	0.53	0.90 (0.65–1.26)	0.54	1.04 (0.74–1.46)	0.82
	12 months (*N* = 1070)	Low [2–6]	563 (52.6)	1		1		1		1		1	
		Middle [7]	256 (24.0)	0.94 (0.66–1.34)	0.75	1.16 (0.74–1.80)	0.50	1.15 (0.67–1.93)	0.59	0.95 (0.69–1.32)	0.76	1.09 (0.79–1.50)	0.62
		High [8]	251 (23.4)	0.78 (0.52–1.13)	0.19	1.34 (0.84–2.10)	0.21	1.28 (0.74–2.19)	0.37	0.90 (0.64–1.27)	0.54	1.02 (0.72–1.45)	0.91

Enamel hypomineralization on at least one permanent tooth (HT ≥ 1); Enamel hypomineralization on at least one first permanent molar (MIH); Enamel hypomineralization on at least one first permanent molar and incisor (M + IH); Significant associations (*p* < 0.05) are written in bold font. Confounders: BMI at age 10; Parental education level as a proxy for socio-economic status, defined as the highest degree achieved either by mother or father (Low for ≤10 years, high for >10 years); Study either GINIplus observation arm, GINIplus intervention arm, or LISA; Maternal atopy defined as mother reported to have hay fever, allergic rhinitis, allergic conjunctivitis, atopic eczema, or physician-diagnosed asthma.

**Table 3 ijerph-18-11411-t003:** Adjusted odds ratios/relative risks with 95% confidence intervals obtained from logistic regression analysis and Poisson hurdle regression analysis of single food items during the first 12 months of life and MIH, M + IH, and number of HTs at 10 years of life. All models were adjusted for sex, study, age, BMI, maternal atopy, and parental education.

Food Groups	Single Food Items	Consumption (%) Yes No		Logistic Model MIH OR (95% CI)	Logistic Model M + IH OR (95% CI)	Hurdle Model HT ≥ 1 Severity RR (95% CI)
Vegetables	Avocado	16	84	1.10 (0.69–1.73)	1.19 (0.68–1.99)	1.39 (0.99–1.96)
	Cauliflower	84	16	1.31 (0.78–2.32)	0.91 (0.52–1.69)	1.25 (0.85–1.83)
	Beans	46	54	1.09 (0.77–1.55)	1.11 (0.73–1.69)	1.06 (0.82–1.38)
	broccoli	82	18	1.15 (0.73–1.89)	0.84 (0.51–1.45)	0.89 (0.64–1.24)
	Peas	77	23	1.62 (1.03–2.63)	1.57 (0.92–2.84)	1.21 (0.88–1.67)
	Cucumber	67	33	1.33 (0.89–2.01)	1.56 (0.96–2.60)	1.09 (0.82–1.46)
	White cabbage	35	65	1.28 (0.88–1.85)	1.47 (0.95–2.27)	1.16 (0.88–1.54)
	Turnip cabbage	54	46	1.14 (0.80–1.64)	1.23 (0.80–1.90)	0.99 (0.76–1.28)
	Lenses	10	90	1.01 (0.53–1.79)	0.59 (0.23–1.30)	0.89 (0.57–1.40)
	Celery	36	64	1.06 (0.73–1.53)	1.16 (0.75–1.78)	0.89 (0.68–1.17)
	Asparagus	24	76	0.76 (0.48–1.17)	0.63 (0.35–1.07)	1.02 (0.75–1.39)
	Spinach	77	23	1.24 (0.82–1.91)	1.16 (0.72–1.94)	1.15 (0.85–1.56)
	Tomatoes	86	14	1.00 (0.61–1.71)	1.13 (0.62–2.24)	1.18 (0.80–1.73)
	Onion	60	40	1.04 (0.72–1.50)	0.86 (0.56–1.32)	1.06 (0.80–1.39)
	Vegetable juice	49	51	1.08 (0.76–1.54)	0.96 (0.63–1.46)	1.01 (0.78–1.32)
Fruit	Pineapples	47	53	1.03 (0.71–1.47)	0.97 (0.63–1.50)	0.95 (0.72–1.23)
	Apricots	71	29	1.59 (1.04–2.50)	2.34 (1.34–4.38)	1.19 (0.88–1.60)
	Strawberries	48	52	0.99 (0.69–1.41)	0.78 (0.51–1.18)	0.92 (0.71–1.20)
	Peaches	75	25	1.12 (0.73–1.74)	1.16 (0.70–1.99)	1.06 (0.77–1.45)
	Citrus fruit	51	49	1.29 (0.90–1.84)	1.37 (0.90–2.10)	1.09 (0.84–1.42)
	Fruit juice	84	16	1.08 (0.68–1.76)	1.04 (0.61–1.86)	1.28 (0.90–1.82)
Cereal	Cookies/cakes/rusk	92	8	0.88 (0.50–1.62)	1.29 (0.63–3.01)	1.00 (0.64–1.58)
	Rolles oats	75	25	1.37 (0.89–2.18)	1.26 (0.76–2.18)	1.23 (0.90–1.68)
	Muesli	37	63	1.18 (0.81–1.69)	1.19 (0.76–2.18)	1.00 (0.76–1.32)
	Millet	28	72	1.26 (0.86–1.83)	1.67 (1.07–2.56)	1.03 (0.77–1.37)
	Cornmeal/corn starch	31	69	1.36 (0.93–1.96)	0.92 (0.57–1.43)	1.11 (0.84–1.46)
	Wheat semolina/starch	71	29	1.58 (1.03–2.50)	1.43 (0.87–2.45)	1.19 (0.88–1.62)
	Spelt	44	56	1.22 (0.85–1.75)	1.50 (0.98–2.30)	1.14 (0.87–1.49)
Meat	Lamb	13	87	1.19 (0.72–1.92)	1.04 (0.55–1.85)	0.80 (0.54–1.18)
	Veal/beef	91	9	1.18 (0.63–2.42)	1.05 (0.52–2.43)	0.79 (0.49–1.25)
	Pork	67	33	0.97 (0.67–1.41)	1.03 (0.67–1.63)	0.95 (0.72–1.25)
	Sausages	70	30	1.12 (0.76–1.68)	1.35 (0.84–2.25)	0.98 (0.73–1.31)
Egg		60	40	1.26 (0.87–1.84)	1.27 (0.82–2.00)	1.11 (0.84–1.45)
Dairy products	Dairy products	84	16	1.49 (0.92–2.50)	1.87 (1.02–3.66)	1.41 (0.98–2.02)
	Cow milk/cream	72	28	1.42 (0.94–2.17)	1.77 (1.07–3.03)	1.31 (0.97–1.76)
	yoghurt/quark/cheese	80	20	1.58 (1.01–2.54)	1.65 (0.97–2.97)	1.33 (0.96–1.85)
Fish		37	63	1.25 (0.85–1.83)	1.27 (0.80–1.99)	0.96 (0.72–1.28)
Others	Others	58	42	1.03 (0.71–1.48)	1.10 (0.72–1.71)	0.90 (0.68–1.17)
	Nuts	24	76	0.84 (0.52–1.31)	0.79 (0.44–1.34)	0.85 (0.61–1.18)
	Soy products	14	86	0.66 (0.35–1.16)	0.49 (0.20–1.01)	0.80 (0.54–1.19)
	Cocoa/chocolate	49	51	1.24 (0.86–1.77)	1.24 (0.81–1.91)	0.99 (0.76–1.30)

Enamel hypomineralization on at least one permanent tooth (HT ≥ 1); Enamel hypomineralization on at least one first permanent molar (MIH); Enamel hypomineralization on at least one first permanent molar and incisor (M + IH); Significant associations according to adjusted alpha level = 0.001; No significant associations (*p* < 0.001) were observed; All analyses were limited to those being consumed by more than 5% and less than 95% of the study population with outcome variables that showed significant associations in the main analyses.

## Data Availability

Due to data protection reasons, the datasets generated and/or analyzed during the current study cannot be made publicly available. The datasets are available to interested researchers from the corresponding author on reasonable request (e.g., reproducibility), provided the release is consistent with the consent given by the GINIplus and LISA study participants. Ethical approval might be obtained for the release and a data transfer agreement from the legal department of the Helmholtz Zentrum München must be accepted.

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
