# Peer review of "Association Study on Nutrition in the First Year of Life and Molar-Incisor Hypomineralization (MIH)—Results from the GINIplus and LISA Birth Cohort Studies"

_ijerph, 2021, doi:10.3390/ijerph182111411_

Round 1

Reviewer 1 Report

Dear Authors,

This follow up on your extensive research is highly welcome and makes a lot of scientific sense. Congratulations on your hard work.

Your results after all these years including such a great number of cases are of course disappointing for both patients as well as clinicians. Your suggestion towards creation of further investigation is definitive substantiated and deserves publication. 

Author Response

Thank you for reviewing and evaluating our work positively!

Reviewer 2 Report

Dear Authors.

Thank you for submitting your paper for publication. This study was well conducted and written up to a high standard. My only concern is in regards to the mention of 'one dentist' as the examiner for 1000+ students across two studies, which seems incorrect. I wish to see further details included on this aspect in the methodology section. 

Regards 

Author Response

Our response:    Thank you for reviewing and evaluating our work! Your comment makes total sense and we changed some of the details. However, there was a team to perform the dental examinations and by “one calibrated dentist”, we meant for each patient, one dentist, and not all of them.

Revised text: Line 95

Reviewer 3 Report

There are sentences where English requires an extensive modification to allow the reader to smoothly follow the content. I suggest the following modifications:
- in the abstract, in line 20: "[...] is a condition with specific clinical presentation which etiology to date still remains unknown". Line 21: remove the term association (in "this association study"). Line 22: I think the correct way to express the association should be this "the association between nutrition during the 1st year of life and the presence of MIH". Line 23: "data from 1070 10-year-old CHILDREN from two prospective birth cohort studies (remove designed)". Line 25: replace "using questionnaires" with "was assessed at 4-, 6-, and 12-month time-points".
- in the introduction: line 59, remove "with respect to this gap in knowledge" and put "therefore, [...]". Line 62: "[...] there is no association BETWEEN early diet in terms of breastfeeding, food item diversity and food group diversity AND MIH development"
- in materials and methods: line 69, replace "earlier" with "elsewhere". 
- in the flow chart of figure 1: remove capital letter to Oral 
- in the results: line 163, "24.9% of the 10-year-old CHILDREN" and the same in line 217
- in the discussion, line 231: replace "when we took a closer look into" with a more formal expression 

Confusing sentence to be rephrased or better explained:

  • in the abstract: line 25, what do you mean for "food diversity was considered according to the number of food items?" do you mean that the definition or the criterion of food diversity was defined as ... Please clarify, and if possible make it more objective.
  • in the abstract: line 29-30, please rephrase. Do you mean that the association between EBF, food item and food group diversity with MIH was not significant or that EBF, food item and food group diversity were not found to be good predictor of the outcome of MIH? I would anyway substitute the "during the first 4-, 6-, 12-months of life" with "at 4-, 6-, 12-month time-points".
  • in the abstract: please rephrase the conclusion (line 31-33), like "despite the limitation of this study (and in that case mention some of them), the results of this study suggest the lack of an association between early nutrition in the first year of life and MIH in the permanent dentition".
  • in the introduction: line 39, rephrase to "Potential related factors have been extensively investigated, but ....".
  • in the material and methods: line 103, how do you define "food diversity"? I appreciate the reference, but I also suggest to provide with a definition (if the definition is the one provided in line 120, I would just move the sentence in line 102 down to the section when you start talking about food diversity, e.g. line 120)
  • please rephrase the sentence in line 121. What does it mean by age 6 and 12 months? Was food diversity calculated according to the N of food they introduced from 12 months to 6 year of age? This is what I understand from the passage. 
  • In material and methods, line 123: please provide the cut-off to be categorized as low-, middle-, and high-diversity groups (indicate them here and not in the results). 
  • in the discussion, line 199-200: the passage is not clear. I would restate it in this way "[...] may lead to defects in enamel; however, it is possible that a duration to 4 months of EBF may not be long enough to detect any visible effect, such in the present study. This is in accordance with ...".
  • in the discussion, line 210: replace with "it may be expected an association between the effect of EBF during the first 4 months ..."  

Passage to be modified:

  • in the introduction (line 38), I suggest to look for literature that also talks about MIH in children, as the article will be focused on 10-yy-old children, and not adolescent.
  • As the sentence in line 42 to line 45 is a list of potential related factors associated with MIH, I would move this list and connected to line 40, when you mention "several potential related factors have been investigated, including ... (etc). However, the etiology of MIH to date has not been clarified"
  • please, clarify in the materials and methods how old the children are when they were examined 
  • in table 1, I would put here the clarification of "maternal atopy", as we encounter this term in table 1 first, and then in table 2. 
  • I would move the clarification of the power analysis to the material and methods or to the statistical analysis section, instead of introducing it in the discussion (line 192-194). 

Some other doubts: how did you obtain a written consent from the participants themselves (line 77)? I would skip this information as it sounds anyway not relevant for a valid consent. Maybe you can indicate that an oral consent was obtained form them, but I have hard time to understand the validity of having 10-yy-old kids sign a consent. 

Which were the missing values (Line 120) in the analysis? Could you simply consider them as randomly occurring? 

What does it mean in statistical analysis (line 155) the confounding of "study group"? Do you mean the level of education"?

What do the numbers in epiphysis indicate in the table 1 after low- middle and high food item diversity? 

I would like to see in the discussion how the confounding factor of sex, BMI, etc may influence the outcome and being considered as confounding factors. 

Author Response

There are sentences where English requires an extensive modification to allow the reader to smoothly follow the content. I suggest the following modifications:

Our response: We appreciate such attention to the details, all the remarks were considered as valuable and we made the necessary changes as following:

- in the abstract, in line 20: "[...] is a condition with specific clinical presentation which etiology to date still remains unknown". Line 21: remove the term association (in "this association study"). Line 22: I think the correct way to express the association should be this "the association between nutrition during the 1st year of life and the presence of MIH". Line 23: "data from 1070 10-year-old CHILDREN from two prospective birth cohort studies (remove designed)". Line 25: replace "using questionnaires" with "was assessed at 4-, 6-, and 12-month time-points".

Revised text:      Line 21 to 25

- in the introduction: line 59, remove "with respect to this gap in knowledge" and put "therefore, [...]". Line 62: "[...] there is no association BETWEEN early diet in terms of breastfeeding, food item diversity and food group diversity AND MIH development"

Revised text:      Line 68 to 72

- in materials and methods: line 69, replace "earlier" with "elsewhere".

Revised text:      Line 78

- in the flow chart of figure 1: remove capital letter to Oral 

Revised text:      Page 3

- in the results: line 163, "24.9% of the 10-year-old CHILDREN" and the same in line 217

Revised text:      Line 185

- in the discussion, line 231: replace "when we took a closer look into" with a more formal expression 

Our response:    Thank you for this remark.

Revised text:      Line XXX

Confusing sentence to be rephrased or better explained:

Our response:    Thank you for this remark. We amended the relevant paragraphs.

in the abstract: line 25, what do you mean for "food diversity was considered according to the number of food items?" do you mean that the definition or the criterion of food diversity was defined as ... Please clarify, and if possible make it more objective.

Our response: Yes, it was the definition so the amendment was done.

Revised text:      Line 26 to 29

in the abstract: line 29-30, please rephrase. Do you mean that the association between EBF, food item and food group diversity with MIH was not significant or that EBF, food item and food group diversity were not found to be good predictor of the outcome of MIH? I would anyway substitute the "during the first 4-, 6-, 12-months of life" with "at 4-, 6-, 12-month time-points".

Our response: We meant the former; that the association between EBF, food item and food group diversity with MIH was not significant.

Revised text:      Line 32 to 34

in the abstract: please rephrase the conclusion (line 31-33), like "despite the limitation of this study (and in that case mention some of them), the results of this study suggest the lack of an association between early nutrition in the first year of life and MIH in the permanent dentition".

Revised text:      Line 34 to 37

in the introduction: line 39, rephrase to "Potential related factors have been extensively investigated, but ....".

Revised text:       Line 46

in the material and methods: line 103, how do you define "food diversity"? I appreciate the reference, but I also suggest to provide with a definition (if the definition is the one provided in line 120, I would just move the sentence in line 102 down to the section when you start talking about food diversity, e.g. line 120)

Revised text:       Line 123

please rephrase the sentence in line 121. What does it mean by age 6 and 12 months? Was food diversity calculated according to the N of food they introduced from 12 months to 6 year of age? This is what I understand from the passage. 

Our response: We tried to make the paragraph as clear as possible by some changes. We summed the number of food items and food groups consumed by the children when they were 6 and 12 months old, then based on their distribution, categorized into three different groups.

Revised text:       Line 132 to 143

In material and methods, line 123: please provide the cut-off to be categorized as low-, middle-, and high-diversity groups (indicate them here and not in the results). 

Our response: We explicitly explained how these categories are made but since they’re different in their ranges, refrained from putting them into the text, instead in the tables.

Revised text:       Line 137 to 138

in the discussion, line 199-200: the passage is not clear. I would restate it in this way "[...] may lead to defects in enamel; however, it is possible that a duration to 4 months of EBF may not be long enough to detect any visible effect, such in the present study. This is in accordance with ...".

Revised text:       Line 222 to 226

in the discussion, line 210: replace with "it may be expected an association between the effect of EBF during the first 4 months ..."  

Revised text:       Line 235 to 236

Passage to be modified:

in the introduction (line 38), I suggest to look for literature that also talks about MIH in children, as the article will be focused on 10-yy-old children, and not adolescent.

Our response: We totally agree with this remark, and the literature stated in the references cover the information for children as well.

Revised text: Line 43

As the sentence in line 42 to line 45 is a list of potential related factors associated with MIH, I would move this list and connected to line 40, when you mention "several potential related factors have been investigated, including ... (etc). However, the etiology of MIH to date has not been clarified"

Our response: changed and all the references are adjusted accordingly.

Revised text: Line 46 to 50.

please, clarify in the materials and methods how old the children are when they were examined 

Revised text: Line 84 and 85

in table 1, I would put here the clarification of "maternal atopy", as we encounter this term in table 1 first, and then in table 2. 

Revised text: Table1

I would move the clarification of the power analysis to the material and methods or to the statistical analysis section, instead of introducing it in the discussion (line 192-194). 

Revised text: Line 176 to 178

Some other doubts: how did you obtain a written consent from the participants themselves (line 77)? I would skip this information as it sounds anyway not relevant for a valid consent. Maybe you can indicate that an oral consent was obtained form them, but I have hard time to understand the validity of having 10-yy-old kids sign a consent. 

Our response: Thank you for pointing this out. We agree that the wording might be misleading. The legally relevant and valid written informed consent was provided by at least one parent or legal guardian. However, the child was asked to co-sign the consent form. To avoid confusion of the reader, we followed the reviewer’s suggestion and changed the sentence.

Revised text: Line 85 to 88

Which were the missing values (Line 120) in the analysis? Could you simply consider them as randomly occurring? 

Our response: Line 133 and 134

Revised text:

What does it mean in statistical analysis (line 155) the confounding of "study group"? Do you mean the level of education"?

Our response: study was either GINIplus observation arm, GINIplus intervention arm, or LISA

Revised text: Line 174

What do the numbers in epiphysis indicate in the table 1 after low- middle and high food item diversity? 

Our response: They indicate the range of the number of food items and food groups which are included in each category. For further clarification, some sentences are added to the methods section.

Revised text: Line 141

I would like to see in the discussion how the confounding factor of sex, BMI, etc may influence the outcome and being considered as confounding factors. 

Our response: Further explanations and the reference are added in the methods section. Also the non significant results are indicated in the result section.

Revised text: Line 175 and Line 208
